# Modeling the formation of social conventions from embodied real-time interactions

**Ismael T. Freire** [1] *, **Clement Moulin-Frier**[2], **Marti Sanchez-Fibla**[3], **Xerxes D. Arsiwalla**[1], **Paul F. M. J. Verschure**[1,4,5]

**1** SPECS Lab, Institute for Bioengineering of Catalonia, Barcelona, Spain, **2** Flowers Team, Inria and Ensta ParisTech, Bordeaux, France, **3** AI-ML Group, ETIC, Universitat Pompeu Fabra, Barcelona, Spain, **4** Barcelona Institute of Science and Technology (BIST), Barcelona, Spain, **5** Catalan Institution for Research and Advanced Studies (ICREA), Barcelona, Spain

* ifreire@ibecbarcelona.eu

**Data Availability Statement:** All relevant data are within the manuscript and its Supporting Information files.

**Funding:** PFMJV. This project has received funding from the European Union's Horizon 2020 research

## Abstract

What is the role of real-time control and learning in the formation of social conventions? To answer this question, we propose a computational model that matches human behavioral data in a social decision-making game that was analyzed both in discrete-time and continuous-time setups. Furthermore, unlike previous approaches, our model takes into account the role of sensorimotor control loops in embodied decision-making scenarios. For this purpose, we introduce the Control-based Reinforcement Learning (CRL) model. CRL is grounded in the Distributed Adaptive Control (DAC) theory of mind and brain, where low-level sensorimotor control is modulated through perceptual and behavioral learning in a layered structure. CRL follows these principles by implementing a feedback control loop handling the agent's reactive behaviors (pre-wired reflexes), along with an Adaptive Layer that uses reinforcement learning to maximize long-term reward. We test our model in a multi-agent game-theoretic task in which coordination must be achieved to find an optimal solution. We show that CRL is able to reach human-level performance on standard game-theoretic metrics such as efficiency in acquiring rewards and fairness in reward distribution.

## Introduction

Our life in society is often determined by social conventions that affect our everyday decisions. From the side of the road we drive on to the way in which we greet each other, we rely on social norms and conventions to regulate these interactions in the interest of group coordination. Seemingly stable, those norms are not carved in stone, but can evolve and change over time, as seen, for instance, in energy consumption patterns of a population [1, 2]. Moreover, small variations in individual behavior driven by social norms can have huge impact on the environment when implemented at a global scale. Among most recent examples of rapidly emerging social norms with unprecedented consequences one might refer to the social distancing convention adopted as one of the major ways to fight the spread of COVID-19.

But what is a convention, how is it formed and maintained over time? Although some forms of cooperative behaviors are observed across different animal and insect species, current

and innovation programme, ID:820742 and ID:641321. MSF and CMF. This project has been supported by INSOCO-DPI2016-80116-P. The funders had no role in study design, data collection and analysis, decision to publish, or preparation of the manuscript.

**Competing interests:** The authors have declared that no competing interests exist.

evolutionary models alone do not seem to provide sufficient explanation of why humans but not other animals exhibit large-scale cooperation among genetically unrelated individuals [3]. Behavioral evidence suggests that "strong reciprocity"—defined as a tendency to voluntarily cooperate, if treated fairly, and to punish non-cooperators—[4] can account for this uniquely human type of cooperation. Such conditional cooperation allows humans to reach conventions, although various groups can differ greatly in particular social norms adopted. This variability in social conventions also serves another function: promoting conformity within groups and heterogeneity across groups [5].

After decades of research on the topic, the boundaries and relationship between the different categories of social norms are still under debate [6]. However, both recent [7] and classic [8] literature do agree on their definition: conventions are patterns of behavior that emerge within a group to solve a repeated coordination problem. More concretely, conventions exhibit two characteristic features [8]: (i) they are self-sustaining and (ii) they are largely arbitrary. Self-sustaining, in the sense that a group of agents in a given population will continue to conform to a particular convention as long as they expect the others to do so; and arbitrary, in the sense that there are other equally plausible solutions to solve the same problem. Identifying the set of conditions that lead to the formation of such conventions is still an open question, traditionally studied through coordination games, a sub-domain of game theory [9] where controlled experimental setups let us gain insights into human behavioral patterns of cooperation and develop hypotheses explaining observed dynamics.

A typical case for a coordination game is the so-called 'Choosing Sides'. This game proposes a situation in which two drivers meet in the middle of a narrow road. Both drivers must make a decision to avoid a fatal collision: whether to turn right or left. If both choose to turn to the same side they will manage to dodge each other, but if they choose differing maneuvers they will collide. The payoff matrix of Table 1 represents numerically this situation.

The solutions to this type of matrix-form game satisfy the two criteria of a convention. First, there are more that one possible solution (both players choose Left or both choose Right), so choosing one or the other is an arbitrary decision. And once a solution is reached, it is more optimal for each player to keep their current decision than to change it unilaterally, so it is also self-sustaining.

Although classical matrix-form games have been extensively investigated in literature over the past decades [10, 11], several studies point to the fact that coordination in realistic social circumstances usually requires a continuous exchange of information in order for conventions to emerge [10, 12, 13]. Precisely, this is a feature that classical matrix-form games lack because they are based on discrete-time turns that impose a significant delay between actions [13–15]. In order to address this problem, recent literature has devised ways to modify standard game theoretic discrete-time tasks into dynamic versions where individuals can respond to the other agent's actions in real or continuous-time [16–21]. Their results point out that cooperation can be more readily achieved in the dynamic version of the task due to the rapid flow of information between individuals and their capacity to react in real-time [17, 22].

A recent example of such an ecological approach can be found in [23], where Hawkins and Goldstone show that continuous-time interactions help to converge to more stable strategies in a coordination game (the Battle of the Exes) compared to the same task modeled in

**Table 1. Choosing sides payoff matrix.**

|  | Left | Right |
|---|---|---|
| Left | 10, 10 | 0, 0 |
| Right | 0, 0 | 10, 10 |

discrete-time. They also show that the involved payoffs affect the formation of social conventions. According to these results, they suggest that real-life coordination problems can be solved either by a) forming a convention or b) through spontaneous coordination. And, critically, the solution depends on what is at stake if the coordination fails. To illustrate this point, they suggest two real-life examples of a coordination problem: On the one hand, when we drive a car, the stakes are high because if we fail to coordinate the outcome could be fatal, so we resort to a convention—e.g. to drive on the right side of the road. On the other hand, when we try to avoid people in a crowded street, we do it "on the fly" because the stakes are low, so it is not risky to rely on purely reactive behaviors (e.g. avoidance behavior) to solve it.

However, despite the substantial amount of behavioral studies and the advances in the development of more ecologically valid setups, a multi-level theoretical account of mechanisms driving coordination is still missing. A substantial step in this direction could be the development of algorithms modelling observed human behaviors. Simulations derived from these algorithms can be used to test predictions and provide insights on the possible mechanisms behind cooperation which cannot be directly validated by behavioral or neurophysiological data. Moreover, these computational models provide us with a fast way of exploring the vast range of possible experimental conditions and as well help us overcome limitations of human studies related to costs of subject participation.

These models can be used at later stages not only for prediction and validation of the existing theories but also for control of artificial intelligent agents. If we aim to integrate robots or intelligent machines into our daily lives, we have to provide them with cognitive models that are able to learn and adapt to our social norms and practices. In order to do that, the algorithms governing the agent must integrate high-level information such as rules and plans with embodied information relevant for acting in the real world.

One fundamental step in this direction will be a model that can account for how lower-level dynamic processes interact with higher-level (strategic) cognitive processes: incorporating real-time components will not only add ecological validity, but also can bootstrap learning through solving the sampling inefficiency problem and reducing the time it takes for the model to achieve acceptable performance levels, often seen as a major drawback of the machine learning methods [24].

In this work we formalize high-level strategic mechanisms as a model-free RL algorithm, and we formalize low-level sensorimotor mechanisms as simple control loops. Finally, we develop a model, called Control-based Reinforcement Learning (CRL), that integrates these two mechanisms as layers in a larger architecture. To do so, we draw upon the Distributed Adaptive Control theory (DAC) [25–27], that proposes that cognition is based on several control layers operating at different levels of abstraction. DAC makes explicit the distinction between real-time control on the one hand (Reactive Layer) and perceptual and behavioral learning on the other hand (Adaptive Layer). It is, therefore, an adequate theoretical framework for understanding the specific roles of these two principles (low-level sensorimotor control and high-level strategic learning) in the formation of social conventions, which is the aim of this paper.

In summary, we introduce a novel two-layered cognitive architecture -CRL- that integrates a low-level reactive control loop to manage within-round conflicts on rapid time scales, along with a policy learning algorithm to acquire across-round strategies over longer time scales. We implement this computational model in embodied cognitive agents involved in a social decision-making task called the Battle of the Exes. We compare performance metrics of the CRL model to results of human behavioral data published in [23]. We run simulations showing that the modeled cognitive agents rely more on high-level strategic mechanisms when the stakes of the game are higher. We also show that low-level sensorimotor control helps enhance

performance in terms of efficiency and fairness. These results provide a computational hypothesis explaining key aspects of the emergence of social conventions such as how cognitive processes operating across different temporal scales interact. Finally, we also provide new experimental predictions to be tested on human behavioral studies.

## Related literature

As for computational modeling of game-theoretical tasks, there is an extensive body of literature where the study of the emergence of conflict and cooperation in agent populations has been addressed, especially through the use of Multi-Agent Reinforcement Learning (for extensive reviews, check [28–30]). In this direction, a lot of focus has been recently directed towards developing enhanced versions of the Deep Q-Learning Network architecture proposed in [31], particularly on their extensions to the social domain [32–35]. This architecture uses a reinforcement learning algorithm that extracts abstract features from raw pixels through a deep convolutional network. Along those lines, some researchers [32–34] are modeling the type of conflicts represented in the classic game-theoretic tasks into more ecologically valid environments [32] where agent learning is based on deep Q-networks [33, 34]. For instance, agents based on this cognitive model are already capable of learning how to play a two-player video game such as Pong from raw sensory data and achieve human-level performance [31], both in cooperative and competitive modes [35]. Other similar approaches have focused on constructing agent models that achieve good outcomes in general-sum games and complex social dilemmas, by focusing on maintaining cooperation [36], by making an agent pro-social (taking into account the other's rewards) [37] or by conditioning its behavior solely on its outcomes [38].

However, in all of the above cases, the games studied involve social dilemmas that only provide one single cooperative equilibrium, whereas the case we study in this paper provides several ones, a prerequisite (arbitrariness) for studying the formation of conventions. Also, the abovementioned examples relax one key assumption of embodied agents, that is, that sensory inputs must be obtained through one's own bodily sensors. Agents in previous studies gather their sensory data from a third person perspective. They are trained using raw pixel data from the screen, in either completely observable [35–38] or partially observable [33, 34] conditions. Another point of difference between previous approaches and the work presented here relates to the continuity of the interaction itself. Most of the work done so far in multi-agent reinforcement learning using game theoretical setups has been modeled using grid-like or discrete conditions [32–34, 36–38]. Although there has been progress insofar as many of these studies provide a spatial and temporal dimension (situatedness) to many classical games, they still lack continuous time properties of real-world interactions.

Still, there are a few recent cases where the coordination task has been modeled in real-time and the agents are situated [35, 39, 40]. However, these models suffer from the so-called sample-inefficiency problem due to the huge amount of episodes they require to reach human level performance. A recent review on Deep Reinforcement Learning [24] points out that one way to solve the sample-inefficiency problem would be to integrate architectural biases that help to bootstrap the learning mechanisms by providing some pre-wired adaptation to the environment the agent will live in. To tackle this issue, the Control-based Reinforcement Learning (CRL) model we introduce in this paper integrates lower-level sensorimotor control loops that help to bootstrap policy learning on the higher level of the cognitive architecture. Moreover, we show that the CRL model is sample efficient, by comparing our results to the experimental human data collected in [23].

In order to do that, in the next section first we begin by describing the benchmark task, a coordination game called the Battle of the Exes [23]. After that, we present the CRL

architecture and its two layers: one dealing with the low-level intrinsic behaviors of the agent and another based on model-free reinforcement learning, allowing the agents to acquire rules for maximizing long-term reward [41]. In the Results section, we first compare the results of our model against the benchmark human data and then we show the contribution of each layer by performing several ablation studies. Finally, we conclude this paper by discussing the main implications of our findings, and also comment on limitations and possible extensions of the current model.

## Materials and methods

### Behavioral benchmark

The Battle of the Exes is a coordination game similar to the classic Battle of the Sexes [42], that imposes the following social scenario: A couple just broke up and they do not want to see each other. Both have their coffee break at the same time, but there are only two coffee shops in the neighborhood: one offers great coffee whereas the other, average coffee. If both go to the great coffee shop they will come across each other and will not enjoy the break at all. Therefore, if they want to enjoy their coffee break, they will have to coordinate in a way that they avoid each other every day. This situation can be modeled within the framework of game theory with a payoff relation such as $a > b > 0$; where $a$ is the payoff for getting the great coffee, $b$ the payoff for the average coffee and $0$ the payoff for both players if they go to the same location.

In [23], Hawkins and Goldstone perform a human behavioral experiment based on the above-mentioned game to investigate how two factors—the continuity of the interaction (ballistic versus dynamic) and the stakes of the interaction (high versus low condition)—affect the formation of conventions in a social decision-making task. Concerning the stakes of the interaction, the payoff matrix is manipulated to create two different conditions: *high* and *low*, based on a bigger and smaller difference between rewards, respectively. The payoff matrices in Fig 1 illustrate these two conditions.

As for the continuity of the interaction, the experiment has a *ballistic* and a *dynamic* condition. In the ballistic condition, as in classical game theory, the players can only choose an action at the beginning of every round of the game, without any further control on the outcome. However, in the dynamic condition, the players can freely change the course of their avatars until one of them reaches a reward (for a visual example of the difference between conditions, check the original videos here). In both conditions, the round ends when one of the players reaches one of the reward spots that represent the coffee shops. Altogether, this results in four conditions: two for the stakes of the interaction (high vs. low) combined with two for the continuity of the interaction (ballistic vs. dynamic). For the experiment, they pair human

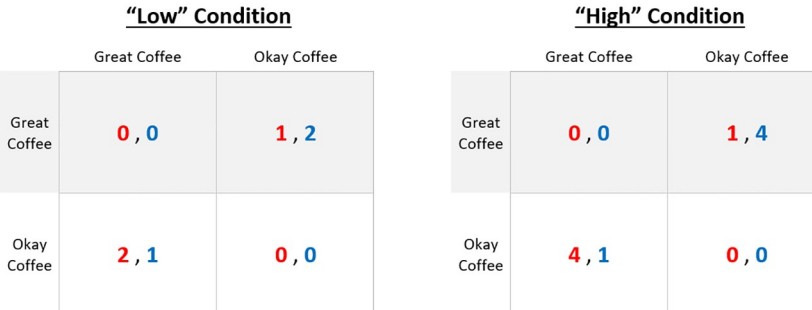

**Fig 1. Payoff matrices of the original "Battle of the Exes" game.** The numbers indicate the reward received by each player (red and blue). Reproduced from [23].

players in *dyads* that depending on the payoff condition, play 50 (high) or 60 (low) consecutive rounds together. In order to analyze the coordination between the players of each dyad, they use three measures -efficiency, fairness, and stability- based on Binmore's three levels of priority [43]:

- *Efficiency*—It measures the cumulative sum of rewards that players were able to earn collectively in each round, divided by the total amount of possible rewards. If the efficiency value is 1, it means that the players got the maximum amount of reward.

- *Fairness*—It quantifies the balance between the earnings of the two players. If the fairness value is 1, it means that both players earned the higher payoff the same amount of times.

- *Stability*—It measures how well the strategy is maintained over time. In other words, it quantifies how predictable the outcomes are of the following rounds based on previous results by *"using the information-theoretic measure of surprisal, which Shannon defined as the negative logarithm of the probability of an event"* [23].

In other words, Efficiency measures utility maximization, Fairness measures the amount of cooperation, and Stability measures the speed and robustness of conventions formed. The results show that players in the dynamic condition achieve greater efficiency and fairness than their counterparts in the ballistic condition, both in the high payoff and low payoff setups. However, their key finding is that in the dynamic condition, the players coordinate more "on the fly" (i.e. without the need of a long-term strategy) when the payoff is low, but when the payoff is high, the participants coordinate into more stable strategies. Namely, they identified the stakes of the interaction as a crucial factor in the formation of social conventions when the interaction happens in real-time.

## Control-based Reinforcement Learning

In this section, we introduce our Control-based Reinforcement Learning (CRL) model. The CRL is composed of two layers, a Reactive Layer and an Adaptive Layer (see Fig 2). The former governs sensorimotor contingencies of the agent within the rounds of the game, whereas the latter is in charge of learning across rounds. This is an operational minimal model, where reinforcement learning interacts with a feedback controller by inhibiting specific reactive behaviors. The CRL is a model-free approach to reinforcement learning, but with the addition of a reactive controller (for model-based adaptive control see [44]).

**Reactive Layer.** The Reactive Layer (RL) represents the agent's sensorimotor control system and is supposed to be pre-wired (typically from evolutionary processes in a biological perspective [24]). In the Battle of the Exes game that we are considering here, we equip agents with two predefined reactive behaviors: *reward seeking* and *collision avoidance*. This means that, even in the absence of any learning process, the agents are intrinsically attracted to the reward spots and avoid collisions between each other. This intrinsic dynamic will bootstrap learning in the Adaptive Layer, as we shall see.

To model this layer, we follow an approach inspired by Valentino Braitenberg's *Vehicles* [45]. These simple vehicles consist of just a set of sensors and actuators (e.g. motors) that, depending on the type of connections created between them, can perform complex behaviors. For a visual depiction of the two behaviors (*reward seeking* and *collision avoidance*), see S1 Video.

- The *reward seeking* behavior is made by a combination of a crossed excitatory connection and a direct inhibitory connection between the reward spot sensors ($s$) and the motors ($m$),

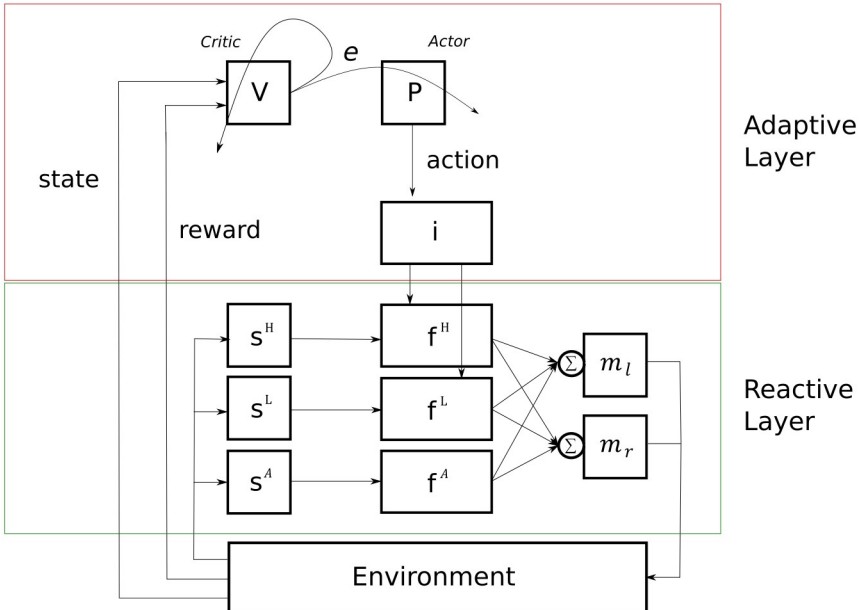

**Fig 2. Representation of the Control-based Reinforcement Learning (CRL) model.** On top, the Adaptive Layer (reinforcement learning control loop) composed of a Critic or value function ($V$), an Actor or action policy ($P$), and an inhibitor function ($i$). At the bottom, the Reactive Layer (sensorimotor control loop), composed of three sets of sensors $s^H, s^L, s^A$, (corresponding to High/Low reward and the other Agent, respectively), three functions $f^H, f^L, f^A$ (corresponding to *High/Low reward seeking* and *collision avoidance* behaviors, respectively) and two motors $m_l, m_r$ (corresponding to the left and right motors). The action selected by the AL is passed through the inhibitor function that will turn off one of the attraction behaviors of the RL depending on the action selected. If the action is *go to the high*, the *low reward seeking* reactive behavior will be inhibited. If the AL selects *go to the low*, the RL will inhibit its *high reward seeking* behavior. If the AL selects *none*, the RL will act normally without any inhibition.

plus a forward speed constant $f$ set to 0.3,

$$m_{left} = f + s^X_{right} - s^X_{left} \tag{1}$$

$$m_{right} = f + s^X_{left} - s^X_{right} \tag{2}$$

where $s^X_{left}$ is the sensor positioned on the left side of the robot indicating the proximity of a reward spot, and $X$ is either the high ($H$) or the low reward ($L$) sensor. The sensors perceive the proximity of the spot. The closer the reward spots, the higher the sensors will be activated. Therefore, if no reward spot is detected ($s^X_{left} = s^X_{right} = 0$), the robot will go forward at speed $f$. Otherwise, the most activated sensor (left or right) will make the robot turn in the direction of the corresponding reward spot.

- The *collision avoidance* behavior is made by the opposite combination: a direct excitatory connection and a crossed inhibitory connection, but in this case between the agent sensors ($s^A$) and the motors ($m$),

$$m_{left} = f + s^A_{left} - s^A_{right} \tag{3}$$

$$m_{right} = f + s^A_{right} - s^A_{left} \tag{4}$$

where $s^A_{left}$ is the sensor positioned on the left side of the robot indicating the proximity of the other agent. The closer the other agent, the higher the sensors will be activated. In this case

as well, if no agent is detected ($s^A_{left} = s^A_{right} = 0$), the robot will go forward at the speed $f$. Otherwise, the most activated sensor will make the robot turn in the opposite direction of the other agent, thus avoiding it.

**Adaptive Layer.** The agent's Adaptive Layer (AL) is based on a model-free reinforcement learning algorithm that endows the agent with learning capacities for maximizing accumulated reward. More specifically, we use an Actor-Critic Temporal Difference Learning algorithm (TD-learning) which is biologically grounded in temporal difference models of animal learning [46–48]. Our implementation is similar to the one described in [49] and is adapted to operate on discrete state and action spaces.

Functionally, it determines the agent's action at the beginning of the round, based on the state of the previous round and its policy. The possible states $s$ are three: *high, low* and *tie*; and they indicate the outcome of the previous round for each agent. That is, if an agent got the high reward on the previous round, the state is *high*; if it got the low reward, the state is *low*; and if both agents went to the same reward, the state is *tie*. The actions $A$ are three as well: *go to the high, go to the low* and *none*. The parameter values reported below have been obtained through a parameter search aimed at fitting the behavioral data in [23].

The TD-learning algorithm is based on the interaction between two main components:

- an *Actor*, or action policy, which learns the mapping from states ($s \in S$) to actions ($a \in A$) and defines what the action ($a$) is, based on a probability ($P$), to be performed in each state ($s$);

$$\pi : S \times A \to [0, 1] \tag{5}$$

$$\pi(a|s) = P(a = a_t|s = s_{t-1}) \tag{6}$$

- and a *Critic*, or value function $V_\pi(s)$, that estimates the expected accumulated reward ($E[R]$) of a state ($s$) following a policy;

$$V_\pi(s_t) = \mathbb{E}[R] = \mathbb{E}\left[\sum_{i=0}^{\infty} \gamma^i r(s_{t+i+1})\right] \tag{7}$$

where $\gamma \in [0, 1]$ is the discount factor, and $r(s_i)$ is the reward at step $i$.

The *Critic* also estimates if the *Actor* performed better or worse than expected, by comparing the observed reward with the prediction of $V_\pi(s)$. This provides a learning signal to the actor for optimizing it, where actions performing better (resp. worse) than expected are reinforced (resp. diminished). This learning signal is called the temporal-difference error (TD error). The TD error $e(s_{t-1})$ is computed as a function of the prediction from value function $V_\pi(s)$ and the currently observed reward of a given state $r(s_t)$,

$$e(s_{t-1}) = r(s_t) + \gamma V_\pi(s_t) - V_\pi(s_{t-1}) \tag{8}$$

where $\gamma$ is a discount factor that is empirically set to 0.40. When $e(s) > 0$ (respectively $e(s) < 0$), this means that the action performed better (resp. worse) than expected. The TD error signal is then sent both to the *Actor* and back to the *Critic* for updating their current values.

The *Critic* (value function) is updated following,

$$V_\pi(s_{t-1}) = V_\pi(s_{t-1}) + \eta e(s_{t-1}) \tag{9}$$

where $\eta$ is a learning rate that is set to 0.15.

The update of the *Actor* is done in two steps. First, a matrix $C(a_t, s_{t-1})$, with rows indexed by discrete actions and columns by discrete states, is updated according to the TD error,

$$C(a_t, s_{t-1}) = C(a_t, s_{t-1}) + \delta e(s_{t-1}) \tag{10}$$

where $\delta$ is a learning rate that is set to 0.45, $a_t$ is the current action and $a_{t-1}$ the previous state. $C(a_t, s_{t-1})$ integrates the observed TD errors when executing the action $a_t$ in the state $a_{t-1}$. It is initialized to 0 for all $a_t$, $a_{t-1}$ and kept to a lower bound of 0. $C(a_t, s_{t-1})$ is then used for updating the probabilities by applying Laplace's Law of Succession [50],

$$P(A = a_t | S = s_{t-1}) = \frac{C(a_t, s_{t-1}) + 1}{\left(\sum_{a \in A} C(a_t, s_{t-1})\right) + k} \tag{11}$$

where $k$ is the number of possible actions.

Laplace's Law of Succession is a generalized histogram (frequency count) where it is assumed that each value has already been observed once prior to any actual observation. By doing so it prevents null probabilities (when no data has been observed, it returns a uniform probability distribution). Therefore, the higher $C(a_t, s_{t-1})$, the more probable $a_t$ will be executed in $a_{t-1}$. Using these equations, actions performing better than expected ($e(s) > 0$) will increase their probability to be chosen the next time the agent will be in state $a_{t-1}$. When $e(s) < 0$, the probability will decrease. If this probability distribution converges for both agents, we consider that a convention has been attained.

The last component of the Adaptive Layer is an inhibitor function, which serves as a top-down control mechanism interfacing between the adaptive and reactive components of the architecture. This function regulates the activity of the reactive control loops based on the action selected by the TD-learning algorithm. In the case that the action selected is *go to the high*, the inhibitor function will shut down the *low reward seeking* behavior, allowing the agent to focus only on the high reward. Conversely, if action were *go to the low*, the reactive attraction to the high reward will be inhibited. Finally, in the case of the *none* action, all reactive behaviors will remain active. For a video showing the Adaptive Layer in action, see S2 Video.

## Results

### Multi-agent simulations

We follow, as in the Battle of the Exes benchmark [23], a 2x2 between-subjects experimental design. One dimension represents the *ballistic* and *dynamic* versions of the game, whereas the other dimension is composed of the *high* and *low* difference between payoffs. Each condition is played by 50 agents who are paired in *dyads* and play together 50 rounds of the game if they are in one of the *high* payoff conditions (ballistic or dynamic), or 60 rounds if they are in one of the *low* payoff conditions. Regarding the task, we have developed the two versions (ballistic and dynamic) of the *Battle of the Exes* in a 2D simulated robotic environment (see Fig 3 for a visual representation). The source code to replicate this experiment is available online at https://github.com/IsmaelTito/CRL-Exes.

In the ballistic condition, there is no possibility of changing the action chosen at the beginning of the round. Therefore, agents only use the Adaptive Layer to operate. The two first actions (high and low) will take the agent directly to the respective reward spots, while the *none* action will choose randomly between them. In each round, the action $a_t$ chosen by the AL is sampled according to $P(A = a_t | S = s_t)$, where $s_t$ is the actual state observed by the agent. In the dynamic condition, however, agents can change their course of action at any point during the round. There the agents use the whole architecture, with the Adaptive Layer and the Reactive Layer working together (see Fig 2).

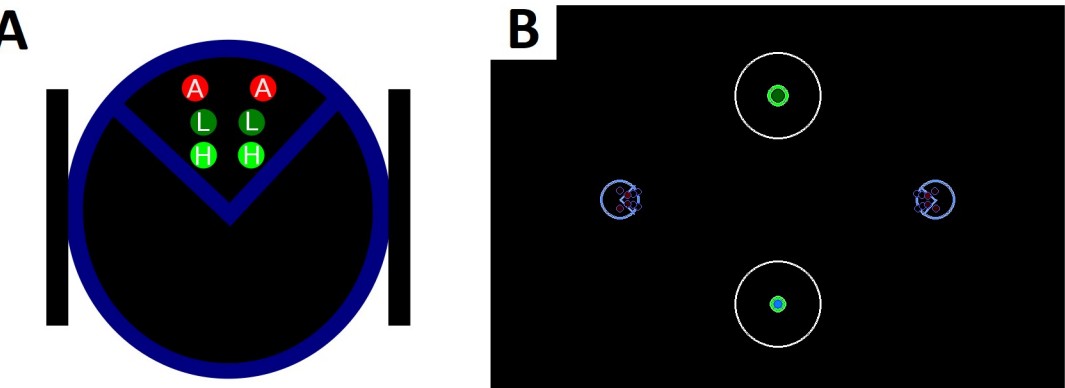

**Fig 3. Experimental setup.** Panel A: Top view of an agent's body, as represented by the dark-blue large circle. Here the agent is facing the top of the page. The two thin black rectangles at the sides represent the two wheels, controlled by their speed. On its front, the agent is equipped with three types of sensors. A: agent sensors (sensing the proximity of the other agent), L: low reward sensors, and H: high reward sensors. For each type, the agent is able to sense the proximity of the corresponding entity both on its left and right side (hence six sensors in total). Panel B: Screenshot of the experimental setup (top view). In blue, the two cognitive agents in their initial position at the start of a round. In green, the two reward spots; the bigger one representing the high reward and the smaller, the low reward (i.e. lower payoff). In white, the circles that delimit the tie area.

The rules of the game are as follows: A round of the game finishes when one of the agents reaches a reward spot. If both agents are within the white circle area when this happens, the result is considered a tie, and both get 0 points. The small spot always gives a reward of 1, whereas the big spot gives 2 or 4 depending on the payoff condition (low or high respectively, see Fig 1). The reward spots are allocated randomly between the two positions at the beginning of each round.

## Model results

We report the main results of our model simulations across the 2x2 conditions, which are analyzed using: efficiency, fairness, and stability [43]. For each of these measures, we first present the results of the model and plot them (see Fig 4, bottom panel) in contrast with human data from [23]. Then we perform a detailed pairwise human-model comparison between conditions, we interpret those results, and finally, we analyze the role of each layer of the CRL architecture.

Regarding the efficiency scores, the results of the CRL model followed a normal distribution in the four conditions, so first, a one-way ANOVA was performed, showing a statistically significant difference between groups ($F(3, 46) = 755$, $p < .001$). Post-hoc independent samples t-tests showed that dynamic (high and low) conditions ($M = 0.88$, $M = 0.86$) are as well significantly more efficient than their ballistic counterparts ($M = 0.46$, $M = 0.45$); between both high ($t(98) = -33$, $p < .001$) and low conditions ($t(98) = -33$, $p < .001$). We observe the same statistical tendencies than in the human benchmark data (see Fig 4, top panel).

As for the fairness scores of the model, a one-way ANOVA showed a statistically significant difference between groups ($F(3, 46) = 6.88$, $p < .001$). Post-hoc independent samples t-tests showed that, as in the benchmark data, the high ballistic condition ($M = 0.50$) is significantly less fair than the two dynamic conditions, high ($t(98) = -3.73$, $p < .001$) and low ($t(98) = -3.99$, $p < .001$), both with a mean fairness score of 0.68. On the other hand, there is no statistically significant difference between the low ballistic condition ($M = 0.61$) and the low dynamic ($p = .086$), but, as opposed to the human results, the low and high ballistic conditions

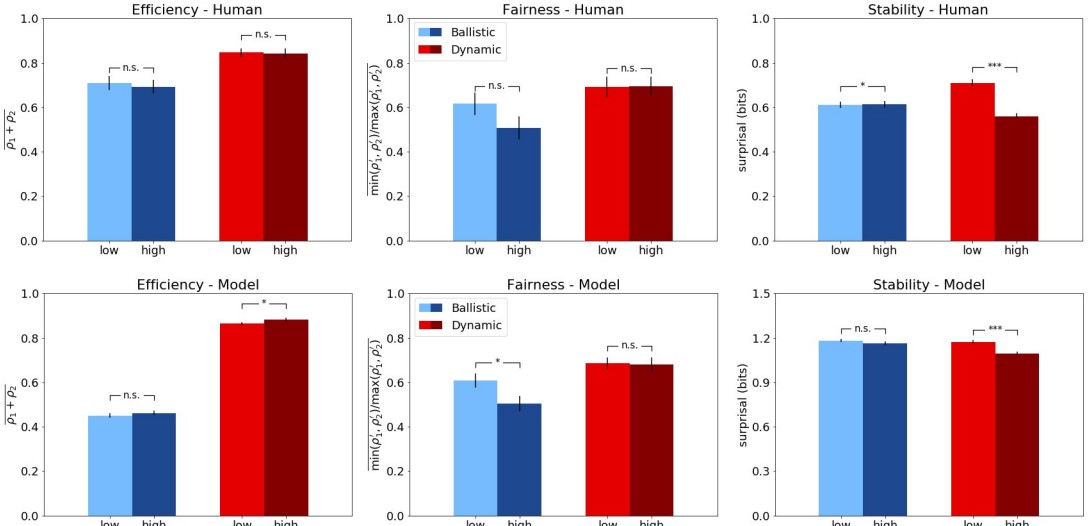

**Fig 4. Results of Control-based Reinforcement Learning compared to human performance in the Battle of the Exes game, measured by efficiency (left), fairness (center) and stability (right).** The top panel shows the human results obtained in [23]. The bottom panel shows the results of the CRL model. Within each panel, blue bars represent the results in the ballistic condition, and red bars represent the results in the dynamic condition. All error bars reflect standard errors.

of the model are significantly different ($t(98) = -2.17$, $p < .05$). Again, the same statistical tendencies are observed in human and model data (see Fig 4, middle panel).

As for stability, since the results of the four conditions showed a non-Gaussian distribution, a non-parametric Kruskal-Wallis H-test was performed, showing a statistically significant difference between groups ($H(3) = 42.3$, $p < .001$). Post-hoc Mann-Whitney U-tests showed that the difference between ballistic low ($M = 1.18$) and dynamic low ($M = 1.17$) conditions is not statistically significant ($p = .10$), but that the dynamic high condition ($M = 1.09$) was significantly more stable than the ballistic high ($M = 1.16$; $p < .001$). There was as well a statistically significant difference between the two dynamic conditions ($p < .001$), but not between the two ballistic ($p = .26$). Similar statistical tendencies are observed in humans and model (see Fig 4, bottom panel).

## Human comparison

Now we perform a pairwise statistical comparison between human and model results to analyse the main points of deviation from the human data. As in the previous section, the results will be compared in terms of Efficiency, Fairness and Stability.

Starting from the efficiency scores on the low-payoff condition (see Fig 5, bottom-left panel), first, normality tests were performed, showing non-normal distributions. Therefore, non-parametric Kruskal-Wallis H-test was performed, showing a statistically significant difference between groups ($H(3) = 98.9$, $p < .001$). Post-hoc Mann-Whitney U-tests showed that there were significant differences in efficiency ($p < .001$) between humans playing the ballistic conditions ($M = 0.70$) of the game and the model ($M = 0.45$). However, there were no significant differences ($p = .34$) between human scores in the dynamic condition ($M = 0.85$) and the scores achieved by the full CRL model ($M = 0.86$). The same statistical relationships are maintained in the high-payoff condition ($H(3) = 102.29$, $p < .001$), where human ballistic scores ($M = 0.69$) and model scores ($M = 0.46$) were significantly different ($p < .001$), while the full

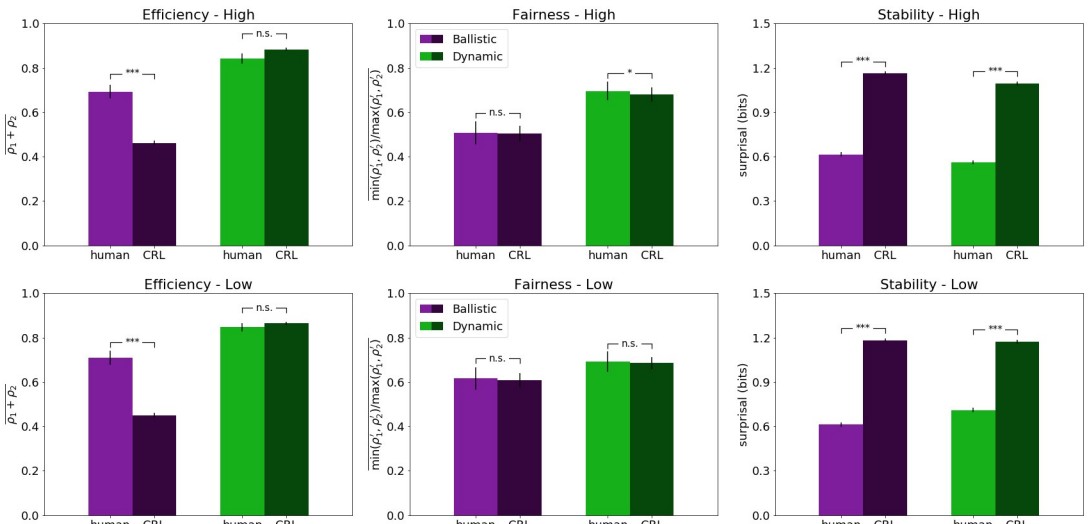

**Fig 5. Pairwise comparison between model and human results in the Battle of the Exes game.** The top panel shows the results on the high-payoff condition. The bottom panel shows the results on the low-payoff condition. Within each panel, blue bars represent the results in the ballistic condition, and red bars represent the results in the dynamic condition. Human data from [23]. All error bars reflect standard errors.

CRL model ($M = 0.88$) shows no statistical difference ($p = .26$) with human dynamic scores ($M = 0.84$).

The Fairness results of the CRL model on the low-payoff condition ($M = 0.61$, $M = 0.69$) matched human scores ($M = 0.61$, $M = 0.69$) in both ballistic and dynamic conditions (see Fig 5, bottom-center panel). In this case the distributions were also non-Gaussian, but the non-parametric Kruskal-Wallis test showed no statistically significant difference between groups ($H(3) = 5.35$, $p > 0.1$). The outcome is similar for the high-payoff condition (see Fig 5, top-center panel). Although this time the Kruskal-Wallis H-test showed a significant difference between groups ($H(3) = 18.74$, $p < .001$), the post-hoc analysis showed no statistical difference ($p = .44$) between human ballistic condition ($M = 0.50$) and model ($M = 0.50$), but significant difference between human dynamic condition ($M = 0.69$) and model ($M = 0.68$, $p = .04$).

On the stability metric, the results of the four conditions showed a non-Gaussian distribution, so a non-parametric Kruskal-Wallis H-test was performed that showed a statistically significant difference between groups ($H(3) = 2385.35$, $p < .001$)). The post-hoc Mann-Whitney U-tests showed that the differences between human ballistic condition ($M = 0.61$) and model ($M = 1.18$), and between human dynamic condition ($M = 0.61$) and model ($M = 1.17$), were statistically significant ($p < .001$ on both cases). On the high-payoff condition, a Kruskal-Wallis also showed significant differences among all stability scores ($H(3) = 2569.62$, $p < .001$). Post-hoc Mann-Whitney U-tests confirmed the statistical difference ($p < .001$) between human ballistic scores ($M = 0.61$) and model ($M = 1.16$). Similarly, human dynamic scores ($M = 0.56$) were significantly smaller ($p < .001$) than the ones obtained by the model ($M = 1.09$).

Overall, the model achieved a good fit with the benchmark data. Like in the human experiment, we observe that the dynamic (real/continuous-time) version of the model achieves better results in efficiency and fairness and that this improvement is consistent regardless of the manipulation of the payoff difference.

The remarkable results in efficiency of the CRL model are due to the key role of the Reactive Layer in avoiding within-round conflict when both agents have chosen to go to the same

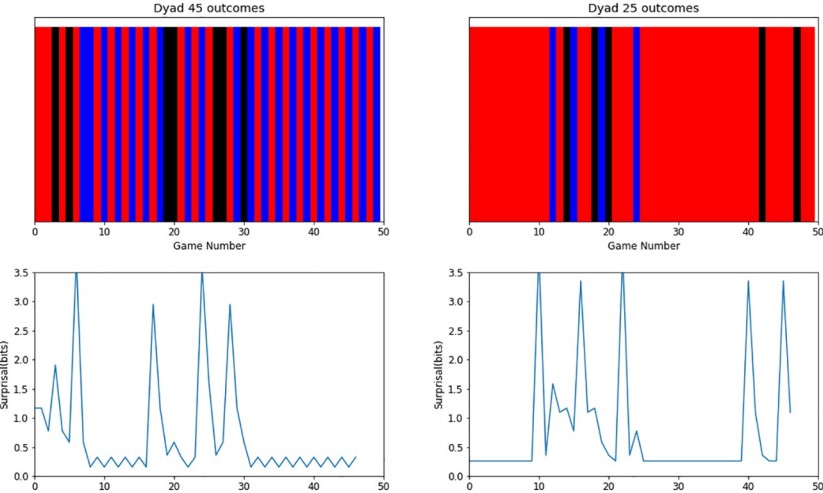

**Fig 6. Dyad results.** Top panel: Outcomes of two dyads of CRL agents (dyad 45 on the left, dyad 25 on the right) in the high dynamic condition, showing the formation of turn-taking (left) and pure dominance (right) equilibria. Each bar represents the outcome of a round of the game. A red bar means that player 1 got the high reward, and a blue bar means that player 2 got the high reward. Black bars represent ties. Bottom panel: Surprisal measure over rounds of play. When a convention is formed, the surprise drops down because the outcomes start to be predictable.

reward, a feature that a ballistic model cannot take into account. The reactive behavior exhibited by the CRL model represents a kind of 'fight or flight' response that can be triggered to make the agent attracted or repulsed to other agents, depending on the context that it finds itself in. In this case, due to the anti-coordination context presented in the Battle of the Exes, the reactive behavior provides the agent with a fast (flight) mechanism to avoid conflict. But in a coordination game like the Battle of the Sexes, this same reactive behavior could be tuned to provide an attraction (fight) response towards the other agent. Future work will extend this model to observe how the manipulation of this reactive behavior can be learned to help the agent in both cooperative and competitive scenarios.

As for the results in stability, the model was overall less stable than the human benchmark data, although it reflected a similar relation between payoff conditions: an increase in stability in the high dynamic condition ($M = 1.09$ and $M = 1.17$) compared to the low dynamic (see Fig 4, right panels). Nonetheless, our results show that social conventions, such as turn-taking and dominance, can be formed by two CRL agents, as shown in Fig 6. The examples shown in the figure illustrate how these two conventions were formed in the dynamic high condition, where these type of equilibria occurred more often and during more rounds than in the other three conditions, thus explaining the higher stability in this condition. Overall, this results are consistent with human data in that dynamic, continuous-time interactions help converge to more efficient, fair and stable strategies when the stakes are high.

## Model comparison

Now we analyze the specific contributions of each layer to the overall results of the CRL architecture. In order to do that, we perform two model-ablation studies, where we compare the results of the whole CRL model against versions of itself operating with only one of its two layers. In the first model ablation, we deactivate the Adaptive Layer, so the resulting behavior of the agents is entirely driven by the Reactive Layer. In the second model ablation, we do the

opposite so the only layer working is the Adaptive Layer. As in the main experiment, there are two payoff conditions (high and low) and 50 dyads per condition.

In terms of efficiency, a one-way ANOVA showed a statistically significant difference between models in both high ($F(2, 47) = 849.47$, $p < .001$) and low conditions ($F(2, 47) = 659.98$, $p < .001$). The post-hoc analysis shows that both the reactive-only ($M = 0.46$, $p < .001$) and adaptive-only models ($M = 0.46$, $p < .001$) obtained a significantly lower score in this metric compared to the whole CRL architecture ($M = 0.88$). This is also true for the low condition (reactive $M = 0.51$, $p < .001$; adaptive $M = 0.45$, $p < .001$; CRL $M = 0.86$). So overall, agents exclusively dependent on one layer perform worse, as we can see in Fig 7 (left panels). This drop in efficiency is caused by a higher amount of rounds that end up in ties, in which both agents do not receive any reward.

On the Fairness metric, the one-way ANOVA also showed statistically significant differences in the two conditions; high ($F(2, 47) = 13.07$, $p < .001$) and low ($F(2, 47) = 5.25$, $p < .01$). In this case, however, the main differences are found among the ablated models, as shown by the post-hoc analysis of both high (reactive $M = 0.71$; adaptive $M = 0.50$, $p < .001$) and low conditions (reactive $M = 0.74$; adaptive $M = 0.61$, $p < .001$). The results in the Fairness score of the ablated models are comparable to the ones of the complete CRL model. However, note that these results are computed from fewer rounds, precisely due to the high amount of ties reached by the ablated models (fairness computes how evenly the high reward is distributed among agents).

Finally, regarding Stability, the normality tests showed non-Gaussian distribution, therefore Kruskal-Wallis non-parametric test was performed, showing significant differences among the three models in the high (($H(2) = 36.82$, $p < .001$) and low (($H(2) = 51.03$, $p < .001$)) conditions. With the post-hoc Mann-Whitney U-tests, we observe that both ablated models are significantly less stable than the CRL model ($M = 1.09$) in the high payoff condition (reactive $M = 1.17$, $p < .001$; adaptive $M = 1.16$, $p < .001$). As for the low condition, the results indicate

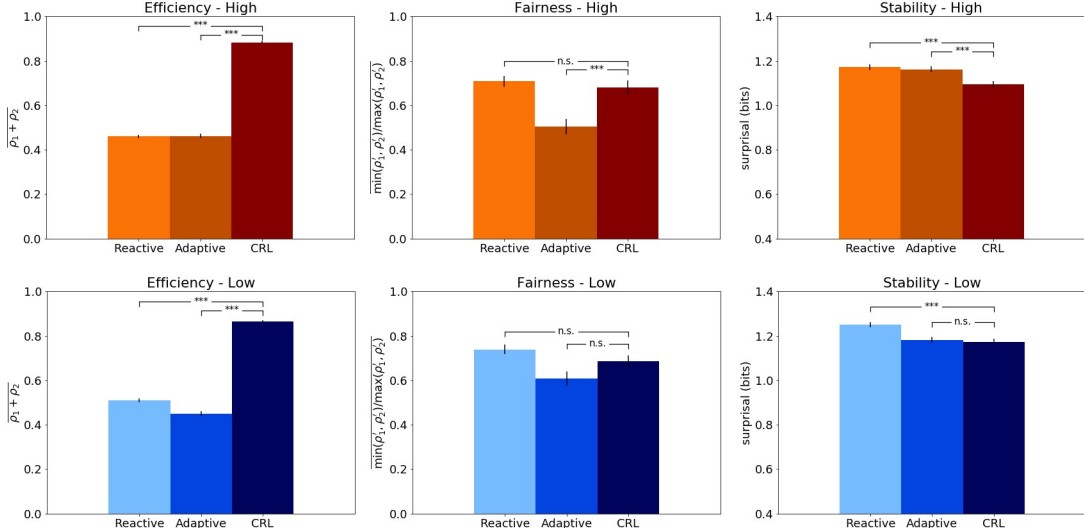

**Fig 7. Model ablation results.** Top panel: Results of the model-ablation experiments compared to the complete CRL results. Orange bars shows the results of the high-payoff conditions, whereas the blue bars refer to the low-payoff conditions. The Reactive model operates using only the Reactive Layer's sensorimotor control. The adaptive-only operates using only the Adaptive Layer's TD-learning algorithm. All results are represented in terms of Efficiency (left panel), Fairness (center) and Stability (right panel). Note that stability is measured by the level of *surprisal*, which means that lower surprise values imply higher stability. All error bars reflect standard errors.

that reactive model ($M$ = 1.25) performs significantly worse in terms of stability when compared to the adaptive ($M$ = 1.18, $p$ < .001) and the CRL models ($M$ = 1.17, $p$ < .001). These results show that overall the ablated models are less stable than the CRL model, as indicated by their higher values in surprise (see Fig 7, right panels). From these model ablation studies we can conclude that any of the layers working alone leads to more unstable and less efficient results.

In a way, these reactive-only and adaptive-only versions of the model instantiate two different approaches of modeling cognition and artificial intelligence [51, 52]. On the one side we have the adaptive-only model implementing a TD-learning algorithm. It represents the symbolic AI tradition, since it works with symbols on a discrete state space [53, 54]. On the other side, the reactive-only model instantiates a pure embodied approach. This model relies only on low-level sensorimotor control loops to guide the behavior of the agent. Therefore, it represents the bottom-up approach to cognition of the behavior-based robotics tradition [55, 56]. But, as we have seen, these models alone are not sufficient to reach human-level performance. Moreover, none of them can match the combination of high-level strategy learning and embodied real-time control shown by the complete CRL model.

## Agents rely more on the Adaptive Layer when stakes are high

We now analyze the participation of each CRL layer across different payoff conditions through the measurement of the "none" action, which refers to the case when the top-down control of the Adaptive Layer is not used during a trial. Based on the results of the benchmark and the CRL model in the dynamic condition, where higher payoff differences helped to achieve higher stability, we expect that the more we increase this difference between payoffs, the more the agents will rely on the Adaptive Layer. For testing this prediction, we have performed a simulation with six different conditions with varying levels of difference between payoffs (high vs. low reward value), from 1-1 to 32-1. To measure the level of reliance on each layer, we logged the number of times each agent outputted a *none* action, that is the action in which the agent relies completely on the Reactive Layer to solve the round.

Considering that there are only 3 possible actions ('go high', 'go low', 'none'), if the Adaptive Layer is randomly choosing the actions, we should observe that the agent selects each action, on average, the same amount of times. That means that prior to any learning, at the beginning of each dyad, the reliance on the Reactive Layer would be 33% and the reliance on the Adaptive Layer 66%. Starting from this point, if our hypothesis is correct, we will expect to observe an increase in the reliance on the Adaptive Layer as the payoff difference increases. As expected, the results confirm, as seen in Fig 8, that there is a steady increase in the percentage of selection of the Adaptive Layer as the payoff difference augments. Such increasing reliance on the Adaptive Layer is possible due to Reactive Layer actively avoiding ties in the early trials, thus facilitating the acquisition of efficient policies.

## Discussion

We have investigated the role of real-time control and learning on the formation of social conventions in a multi-agent game-theoretic task. Based on principles of distributed adaptive control theory, we have introduced a new Control-based Reinforcement Learning (CRL) cognitive architecture. The CRL model uses a model-free approach to reinforcement learning, but with the addition of a reactive controller. The CRL architecture is composed of a module based on an actor-critic TD learning algorithm that endows the agent with learning capacities for maximizing long-term reward, and a low-level sensorimotor control loop handling the agent's reactive behaviors. This integrated cognitive architecture is applied to a multi-agent

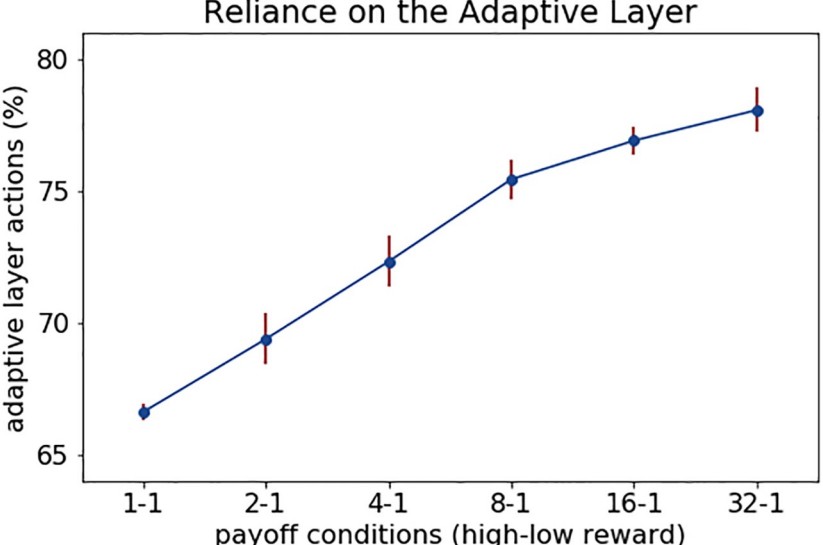

**Fig 8. Reliance on the Adaptive Layer.** Mean of the percentage of Adaptive Layer actions (ie. *go to the high* and *go to the low* actions) selected by the agents plotted against 6 conditions with an increasing difference between high and low payoffs. Bars reflect standard errors.

game-theoretic task, the *Battle of the Exes*, in which coordination between two agents can be achieved. We have demonstrated that real-time agent interaction does affect the formation of more stable, fair and effective social conventions when compared to the same task modeled in discrete-time. The results of our model are consistent with those of Hawkins and Goldstone obtained with human subjects in [23].

Interpreting our results in the context of a functional cognitive model we have elucidated the role of reactive and adaptive control loops in the formation of social conventions and of spontaneous coordination. We found that the Reactive Layer plays a significant role in avoiding within-round conflict (spontaneous coordination), whereas the Adaptive Layer is required to achieve across-round coordination (social conventions). In addition, the CRL model supports our hypothesis that higher payoff differences will increase the reliance on the Adaptive Layer. Based on the differences obtained between the ballistic and dynamic conditions, our results might also suggest that initial successful interactions solved primarily by reactive sensorimotor control can speed up the formation of social conventions.

In our simulations, we have also modeled extensions of experimental conditions (such as increasing differences between payoffs, presented in Fig 8) which affect task outcomes as well as functionality of each control loop. These results allow us to make predictions that can later be tested in new human experiments. More concretely, based on our simulations, we predict that an increased difference in value between the two rewards will promote a faster convergence towards a convention in cooperation games such as The Battle of the Exes. At the cognitive level we suggest that this increase in convention formation could be linked to a higher level of top-down cognitive control, as predicted by the increase in activation of the Adaptive Layer of the CRL model.

Furthermore, there is a biological correspondence of the functions identified by modules of the CRL architecture. Computations described by temporal difference learning are related to the temporal difference model of animal learning [46–48] and have been found in the human brain, particularly in the ventral striatum and the orbitofrontal cortex [57]. It has also been

shown that premotor neurons directly regulate sympathetic nervous system responses such as fight-or-flight [58]. The top-down control system of the brain has been identified in the dorsal posterior parietal and frontal cortex, and shown to be involved in cognitive selection of sensory information and responses. On the other hand, the bottom-up feedback system is linked to the right temporoparietal and ventral frontal cortex and is activated when behaviorally relevant sensory events are detected [59–61].

To the best of our knowledge, this is the first embodied and situated cognitive model that is able to match human behavioral data in a social decision-making game in continuous-time setups. Moreover, unlike previous attempts, we take into account the role of sensorimotor control loops in solving social coordination problems in real-life scenarios. This is arguably a fundamental requirement for the development of a fully embodied and situated AI.

Regarding the limits of the model, we observe that the CRL model still does not reach human level performance in terms of stability. Although the model is quite sample-efficient and reaches good performance in very few trials, it is clear that it does not learn at the same rate as humans. This is because the model-free algorithm implemented at the Adaptive Layer obviously does not capture the cognitive complexity of strategic human behavior. For instance, people can inductively abstract to a 'turn-taking' strategy while the Adaptive Layer would have to separately learn policies for the 'up' and 'down' states from scratch. This can be clearly seen when the model is compared to human performance in the ballistic conditions, where only the Adaptive Layer is active. It is also worth mentioning that in the current paper the model has only been tested in a game where payoffs are fixed, therefore another challenge would be implementing a scenario where payoffs are uncertain. This has been shown to affect cooperation in behavioral experiments [62].

In addition, the CRL model represents a very minimally social model of convention formation (almost as minimal as the naive RL model of [63]). The only way in which the existence of other agents is incorporated into decision-making is through the Reactive Layer's avoidance mechanism, since its modulated by the presence of the other agent. At most, since the state variable (s) depends on the actions selected by each agent in the previous round of the game, one could argue that this variable implicitly takes into account information about the opponent. Besides that, the agents are agnostic to where the rewards are coming from, and certainly not representing and updating the other agents' latent policy or engaging in any kind of social cognition when planning.

But, how much cognitive sophistication is really needed to find solutions to social coordination problems? Our results suggest that we do not need to invoke any advanced social reasoning capabilities for achieving successful embodied social coordination. At least, on this type of coordination problems. Arguably in more complex social scenarios (e.g. generalizing or transferring conventions from one environment to another) a certain level of social representation may become necessary.

For future work, there are several directions in which we can continue to develop research presented in this paper. One obvious extension would be to make the CRL model social. This could be done by representing its partner as an intentional agent or trying to predict and learn what its partner is going to do (as inverse RL, or Bayesian convention formation theories do). Such extension would try to account for individual fixed effects observed in this type of game-theoretical scenarios where different levels of recursive reasoning have been reported [64].

In this paper we have chosen a TD-learning algorithm as it is considered a validated model of animal learning, tuning its parameters through a search that allowed us to get best fit to data. But the CRL architecture allows for the implementation of other algorithms, as the one shown in an extension of this work [65] that integrates loss aversion bias into a Q-learning

algorithm. Therefore, a fruitful avenue for further research would be a more in-depth exploration of the algorithmic and parameter space of the Adaptive Layer.

Another possibility is the addition of a memory module to the CRL architecture. As discussed in [26, 41], this will facilitate the integration of sensory-motor contingencies into a long-term memory that allows for learning of rules. This is important for building causal models of the world and taking into account context in the learning of optimal action policies. The goal of such extensions can be to build meta-learning mechanisms that can identify the particular social scenario in which an agent is placed (i.e., social dilemmas, coordination problems, etc.) and then learn the appropriate policy for each context. Extending our model with such functionality could enable solving more diverse and complicated social coordination problems, both at the dyadic and at the population levels.

Lastly, another interesting avenue concerns the emergence of communication. We could extend our model by adding signaling behaviors to agents and testing them in experimental setups similar to the seminal sender-receiver games proposed by Lewis [8]. One could also follow a more robot-centric approach such as that of [66, 67]. These approaches enable one to study the emergence of complex communicative systems embedding a proto-syntax [49, 68].

Put together, our model in this paper along with recent related work (see [52]) helps towards advancing our understanding of how a functional embodied and situated AI that can operate in a multi-agent social environment. For this purpose, we plan to extend this model to study other aspects of cooperation such as in wolf-pack hunting behavior [69, 70], and also aspects of competition within agent populations as in predator-prey scenarios. In ongoing work, we are developing a setup in which embodied cognitive agents will have to compete for limited resources in complex multi-agent environments. This setup will also allow us to test the hypothesis proposed in [71–73] concerning the role of consciousness as an evolutionary game-theoretic strategy that might have resulted through natural selection triggered by a cognitive arms-race between goal-oriented agents competing for limited resources in a social world.

## Supporting information

**S1 Video. Example of reactive behaviors.** In this example the agents use only the Reactive Layer, not the complete CRL architecture, to play the Battle of the Exes game.
(MP4)

**S2 Video. Example of reactive + adaptive behaviors.** Examples of two agents interacting on the Battle of the Exes game with the two layers of the CRL architecture active.
(MP4)

## Author Contributions

**Conceptualization:** Ismael T. Freire, Clement Moulin-Frier, Marti Sanchez-Fibla, Xerxes D. Arsiwalla.

**Formal analysis:** Ismael T. Freire.

**Funding acquisition:** Paul F. M. J. Verschure.

**Investigation:** Ismael T. Freire.

**Methodology:** Ismael T. Freire.

**Project administration:** Paul F. M. J. Verschure.

**Resources:** Ismael T. Freire, Paul F. M. J. Verschure.

**Software:** Ismael T. Freire.

**Supervision:** Clement Moulin-Frier, Marti Sanchez-Fibla, Xerxes D. Arsiwalla, Paul F. M. J. Verschure.

**Validation:** Clement Moulin-Frier, Marti Sanchez-Fibla, Xerxes D. Arsiwalla, Paul F. M. J. Verschure.

**Visualization:** Ismael T. Freire.

**Writing – original draft:** Ismael T. Freire.

**Writing – review & editing:** Ismael T. Freire, Clement Moulin-Frier, Marti Sanchez-Fibla, Xerxes D. Arsiwalla, Paul F. M. J. Verschure.

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
