## [Decision Letter · Decision Letter 0]

13 Feb 2020

PONE-D-19-36061

Modeling the Formation of Social Conventions from Embodied Real-Time Interactions

PLOS ONE

Dear Mr. Freire González,

Thank you for submitting your manuscript to PLOS ONE. After careful consideration, we feel that it has merit but does not fully meet PLOS ONE’s publication criteria as it currently stands. Therefore, we invite you to submit a revised version of the manuscript that addresses the points raised during the review process.

Please account for the reviever 2's comments on statistical analysis and comparisons of the human/non-human conditions - mainly the reviewer's comment:

"This requires comparing *model* behavior across the 2x2 conditions (if I recall, this was done correctly in the previous submission I reviewed, and I wasn't sure why it was changed here.) Given this first-order analysis, which is currently missing, the model-human comparisons currently reported could serve as an illuminating follow-up analysis exposing systematic points of deviation from the human data (i.e. motivating discussion of why the model has such poorer stability overall, and poorer efficiency in the *ballistic* condition, where humans still form conventions quite readily?)"

Also, account for the specific comments of Review on how to reorganize the material for better understanding. Reviewer 1 has here useful suggestions on clarifying the motivation of your paper for general audience, for example by adding examples, and including relevant literature on social norms. Please follow these suggestions as well.

Also, correct non-working gitlab link(s).

We would appreciate receiving your revised manuscript by Mar 29 2020 11:59PM. To enhance the reproducibility of your results, we recommend that if applicable you deposit your laboratory protocols in protocols.io, where a protocol can be assigned its own identifier (DOI) such that it can be cited independently in the future. For instructions see: http://journals.plos.org/plosone/s/submission-guidelines#loc-laboratory-protocols

We look forward to receiving your revised manuscript.

Kind regards,

Jana Vyrastekova

Academic Editor

PLOS ONE

Journal Requirements:

Reviewers' comments:

Reviewer's Responses to Questions

**Comments to the Author**

1. Is the manuscript technically sound, and do the data support the conclusions?

Reviewer #1: Yes

Reviewer #2: Yes

2. Has the statistical analysis been performed appropriately and rigorously? 

Reviewer #1: Yes

Reviewer #2: No

3. Have the authors made all data underlying the findings in their manuscript fully available?

Reviewer #1: Yes

Reviewer #2: No

4. Is the manuscript presented in an intelligible fashion and written in standard English?

Reviewer #1: Yes

Reviewer #2: Yes

5. Review Comments to the Author

Reviewer #1: The paper studies the role real time control and learning play in the formation of social conventions by using different computational models (simulations) and compare the outcome of these simulations with data from human interactions.

The authors show that in order to reach stable – across round - coordination an adaptive layer in the algorithm is required.

The paper is well written, methodologically sound and makes an interesting contribution. I summarize minor comments (in no particular order) below:

1. Regarding the motivation and literature on social norms and human behaviour, I believe you are missing several papers (summarized in under literature). These papers could help to better motivate the paper with respect to how human cooperation and social norms evolve over time (see, e.g., Cialdini and Trost (1998), Fehr et al. (2002), Fehr and Fischbacher (2003,2004) – which is, I believe, important to know before studying how computer algorithms may mimic (or even outperform) human behaviour. This literature may also help to give examples in real life where social norms play an important role (see, e.g., the environmental domain (Allcott (2011) and Brülisauer et al. (2018)).

2. The motivation of the paper could be more clear. This is a general interest journal and you might have readers who are also from other disciplines and not familiar with the literature. In this regard, I would like to see a better motivation of why your research is important. It is clear that human cooperation is important. It is also obvious that algorithms play an important role – why, however, should we be interested in algorithms that mimic human behaviour? To me the your contribution would be to better understand human corporation per se but it is not so clear in the current version of the manuscript. It is also not clear why we would need an algorithm for this? What can we learn from the algorithm that we cannot learn from experiments with humans (in the lab or in the fmri, for example). It is also not clear why exactly “a model that can account for how lower-level dynamic processes interact with higher-level (strategic) cognitive processes” is exactly the next step needed. It would be good if the authors could elaborate more on this point. Moreover, I would like to see where such algorithms may be applied in the real world – if there is any application. Furthermore, why would we be interested in algorithms (for application) that are similar to humans? Should we not compose algorithms that outperform humans with regard to fairness and efficiency, too?

3. I would also like to see a better motivation of the restrictions applied to the different algorithms. The TD, for example, focusses on long term reward. Would incorporating an algorithm that solely focusses on short term reward change results (I know the focus is coordination over time but there is a substantial literature showing that humans often act irrationally and focus on the present rather than thinking long term and considering future interactions/outcomes (e.g., present bias). I am not asking for such simulations – but I would like to have more focussed discussion on the properties of the algorithms and what might change if they have different properties or, e.g., why considering different properties is of minor importance.

4. Since I am not from the field, I do not want to/and cannot judge the implication of the models itself. What I am missing though is a clear explanation why the TD is only applied in the ballistic setting. Additionally, would it not make sense to compare the models directly, too.

5. Statistical testing: you use non-parametric test throughout the paper (and present no test in Section 3.1. but refer to statistical significance of results.

I would appreciate if you could show the robustness of the results also in parametric regressions (can you account for individual fixed effects when using the human data?). Moreover, please also provide results for the test performed in Section 3.1.

6. Discussion: The discussion should be a bit more broad. In the paper, the authors concentrate on a very specific game – which is important – but also consider certainty in payoffs. In many situations, however, payoffs are uncertain which has consequences for cooperation (see, e.g., Xiao and Kunreuther, (2016)). It would be desirable if you could at least mention this as a further limitation of your algorithms which work only under certainty of payoffs.

Literature

Cialdini, R. B., & Trost, M. R. (1998). Social influence: Social norms, conformity and compliance. In D. T. Gilbert, S. T. Fiske, & G. Lindzey (Eds.), The handbook of social psychology (p. 151–192). McGraw-Hill.

Fehr, E., Fischbacher, U., & Gächter, S. (2002). Strong reciprocity, human cooperation, and the enforcement of social norms. Human nature, 13(1), 1-25.

Fehr, E., & Fischbacher, U. (2004). Social norms and human cooperation. Trends in cognitive sciences, 8(4), 185-190.

Fehr, E., & Fischbacher, U. (2003). The nature of human altruism. Nature, 425(6960), 785-791.

Allcott, H. (2011). Social norms and energy conservation. Journal of public Economics, 95(9-10), 1082-1095.

Brülisauer, M., Goette, L., Jiang, Z., Schmitz, J., & Schubert, R. (2018). Appliance Specific Feedback and Social Comparisons: Evidence From a Field Experiment on Electricity Saving. Working Paper

Xiao, E., & Kunreuther, H. (2016). Punishment and cooperation in stochastic social dilemmas. Journal of Conflict Resolution, 60(4), 670-693.

Reviewer #2: The authors present a model of how agents coordinate in real time, which integrates a globally adaptive, TD-learning algorithm with a locally reactive, embodied mechanism. This model is evaluated on continuous-time and discrete-time versions of a benchmark task that has posed a challenge for classical models relying purely on game theoretic machinery. The model is shown to successfully account for patterns of convention formation as a function of the 'stakes' encoded in the game's payoffs by learning to rely more or less on the adaptive layer, and ablation studies show that both 'strategic' and 'embodied' components are needed.

This is highly original modeling work that should be of great interest and impact to multiple research communities, and I highly recommend acceptance. I’ve been invited to review an earlier iteration of this work, and while I felt quite positive about the paper in that previous cycle, I appreciate that the authors have considerably strengthened the presentation for this submission. The new introduction, in particular, does a great job of cutting to the heart of the contribution, and I greatly appreciate the inclusion of a full set of ablation studies to clarify the contributions of the different architecture components.

In addition to the relatively minor suggestions noted below, I think the manuscript would be improved by one more significant revision: I'm concerned that the results in Section 3 are not reporting the relevant comparisons. Currently, all of the tests are directly pairwise between human and model at each factorial condition, where a significant effect is taken as evidence of a mismatch, and a non-significant effect is taken as evidence that the model is matching human scores. Under the logic of null-hypothesis testing, however, a failure to reject the null hypothesis should not be taken as evidence in favor of the null hypothesis, and any of these numbers could hypothetically be made significant by increasing the number of simulations to a sufficiently high N.

In my view, the first-order question of interest is *not* whether the model gets exactly the same numeric scores as humans for each condition but whether it produces the same qualitative pattern *across* conditions, i.e. whether the model has higher efficiency/fairness in the dynamic condition than the ballistic condition, and whether there is an interaction with payoff for stability. This requires comparing *model* behavior across the 2x2 conditions (if I recall, this was done correctly in the previous submission I reviewed, and I wasn't sure why it was changed here.) Given this first-order analysis, which is currently missing, the model-human comparisons currently reported could serve as an illuminating follow-up analysis exposing systematic points of deviation from the human data (i.e. motivating discussion of why the model has such poorer stability overall, and poorer efficiency in the *ballistic* condition, where humans still form conventions quite readily?)

Specific comments:

* the ‘inhibitor’ function i connecting the two layers is a key component of the proposed architecture but it is not explicitly described in section 2.2 of the main text (only in the caption of Fig. 2 and informally in the "multi-agent simulations" section). Presumably the mechanism connecting the layers is a general feature of the architecture, not just a minor detail of the simulations -- this is also important for understanding the "agents rely more on the adaptive layer when stakes are high" result. I would like to see the description buried in the "In the dynamic condition..." paragraph of section 2.3 moved to the main architecture presentation in 2.2 (possibly as a new subsection 2.2.3, or just as part of the adaptive layer section).

* Again, as an organizational idea, I might move section 2.3 (with the details of the simulation setups) to the beginning of section 3, to make a cleaner distinction between the general CRL formulation and the specifics of the simulations. When reading the results, I was finding myself confused about, for example, how many simulated games the model statistics were based on.

* It would be helpful to include some supplementary videos of a few games played by agents, to give an interested reader a more qualitative sense of their behavior in various conditions (e.g. how collision avoidance and reward seeking interact when purely using the reactive-layer, how the adaptive-layer modifies this behavior, etc.)

* Fig. 6 would be easier to read if (a) the facets corresponded to Fig. 4, where the low- and high- were split out into the top and bottom rows, (b) each facet had exactly three bars, showing CRL, Adaptive-only, and Reactive-only, to quickly read off the 'contribution' of each (currently the reader has to go back and forth between the y-axis of the top and bottom row to judge which of the lesions had more of an impact on efficiency.

* I was interested in how the 'reliance on adaptive layer' shown in Fig. 7 is learned over the course of a game; presumably the model gradually learns on early rounds that a 'none' action leads to a higher payoff on average? Is there any evidence of path-dependence, where the reactive-layer 'bootstraps' conventions, avoiding long runs of ties in early rounds? Perhaps a couple sentences could be added to clarify the mechanism giving rise to the reliance?

* The gitlab link in the paper seems to be broken (gitlab.com/specslab gave a 404 error)

* Typo: "Although there has been progress insofar many …" (should be “insofar as many”)

* Typo: "avoid colliding between each other" (should be either “avoid collisions between” or “avoid colliding into each other”)

6. PLOS authors have the option to publish the peer review history of their article (what does this mean?). If published, this will include your full peer review and any attached files.

Reviewer #1: No

Reviewer #2: Yes

---

## [Author Response · Author response to Decision Letter 0]

21 Apr 2020

Dear editor and reviewers,

First of all, thank you for your time and efforts to review this manuscript. We truly appreciate the feedback which indeed helped us to improve the quality of the paper. As required, the text of the manuscript has been amended to address the all the points raised (see separate submission with changes highlighted). The attached document (Response to Reviewers.pdf) is divided in two sections. In the first one we provide a global response to the main concerns identified by the academic editor. In the second one we provide a detailed answer to each of the reviewer’s comments. We hope that the revision is satisfactory and the paper would now be found suitable for publication. Please do not hesitate to contact us with any further questions or queries.

In the reply below, we adopt the following formatting conventions:

- Reviewer’s comments are in bold,

- Author’s replies are in normal text

- Extracts from the revised version are in italics, indented right.

---

## [Decision Letter · Decision Letter 1]

27 May 2020

Modeling the Formation of Social Conventions from Embodied Real-Time Interactions

PONE-D-19-36061R1

Dear Dr. Freire González,

We are pleased to inform you that your manuscript has been judged scientifically suitable for publication and will be formally accepted for publication once it complies with all outstanding technical requirements.

With kind regards,

Carlos Gracia-Lázaro

Academic Editor

PLOS ONE

Additional Editor Comments (optional):

Reviewers' comments:

Reviewer's Responses to Questions

**Comments to the Author**

1. If the authors have adequately addressed your comments raised in a previous round of review and you feel that this manuscript is now acceptable for publication, you may indicate that here to bypass the “Comments to the Author” section, enter your conflict of interest statement in the “Confidential to Editor” section, and submit your "Accept" recommendation.

Reviewer #1: All comments have been addressed

Reviewer #2: All comments have been addressed

2. Is the manuscript technically sound, and do the data support the conclusions?

Reviewer #1: Yes

Reviewer #2: Yes

3. Has the statistical analysis been performed appropriately and rigorously? 

Reviewer #1: Yes

Reviewer #2: Yes

4. Have the authors made all data underlying the findings in their manuscript fully available?

Reviewer #1: Yes

Reviewer #2: Yes

5. Is the manuscript presented in an intelligible fashion and written in standard English?

Reviewer #1: Yes

Reviewer #2: Yes

6. Review Comments to the Author

Reviewer #1: Dear authors,

You did a good job addressing all my comments.

I believe the paper to be drastically improved compared with the previous version.

Kind regards

Reviewer #2: (No Response)

7. PLOS authors have the option to publish the peer review history of their article (what does this mean?). If published, this will include your full peer review and any attached files.

Reviewer #1: No

Reviewer #2: Yes: Robert Hawkins

---

## [Editor Report · Acceptance letter]

10 Jun 2020

PONE-D-19-36061R1 

Modeling the Formation of Social Conventions from Embodied Real-Time Interactions 

Dear Dr. Freire:

I'm pleased to inform you that your manuscript has been deemed suitable for publication in PLOS ONE. Congratulations! Your manuscript is now with our production department. 

Kind regards, 

on behalf of

Dr. Carlos Gracia-Lázaro 

Academic Editor

PLOS ONE